# Development of Coumarin-Based Hydroxamates as Histone Deacetylase Inhibitors with Antitumor Activities

**DOI:** 10.3390/molecules25030717

**Published:** 2020-02-07

**Authors:** Na Zhao, Feifei Yang, Lina Han, Yuhua Qu, Di Ge, Hua Zhang

**Affiliations:** 1School of Chemistry and Chemical Engineering, University of Jinan, Jinan 250022, China; zhaona3702@163.com; 2School of Biological Science and Technology, University of Jinan, Jinan 250022, China; hanln95@163.com (L.H.); 17854175513@163.com (Y.Q.); gedi_blue@126.com (D.G.)

**Keywords:** coumarin, anti-tumor, HDAC inhibitors, structural modification, anti-proliferative

## Abstract

Histone deacetylases (HDACs) have been proved to be promising targets for the treatment of cancer, and five histone deacetylase inhibitors (HDACis) have been approved on the market for the treatment of different lymphomas. In our previous work, we designed a series of novel coumarin-containing hydroxamate HDACis, among which compounds **6** and **7** displayed promising activities against tumor growth. Based on a molecular docking study, we further developed 26 additional analogues with the aim to improve activity of designed compounds. Several of these new derivatives not only showed excellent HDAC1 inhibitory effects, but also displayed significant growth inhibitory activities against four human cancer cell lines. Representative compounds, **13a** and **13c**, showed potent anti-proliferative activities against solid tumor cell lines with IC_50_ values of 0.36–2.91 µM and low cytotoxicity against Beas-2B and L-02 normal cells. Immunoblot analysis revealed that **13a** and **13c** dose-dependently increased the acetylation of histone H3 and H4. Importantly, the two compounds displayed much better anti-metastatic effects than SAHA against the MDA-MB-231 cell line. Moreover, **13a** and **13c** arrested MDA-MB-231 cells at G2/M phase and induced MDA-MB-231 cell apoptosis. Finally, the molecular docking study rationalized the high potency of compound **13c**.

## 1. Introduction

Cancer is one of the most mortal diseases in the world [1]. In addition to genetic factors, the occurrence of cancer also involves epigenetic modifications including covalent modifications of DNA (methylation and demethylation) and histones [2,3,4,5,6]. Epigenetic regulations achieve reversible modification process through the corresponding enzymes. Histone lysine acetylation level, regulated by histone deacetylases (HDACs) and histone acetyl-transferases (HATs), plays a key role in epigenetic modification [7,8,9,10,11]. Overexpression of HDACs have been found in different cancers [12,13], and histone deacetylase inhibitors (HDACis) have been shown to significantly suppress cell proliferation, angiogenesis and metastasis through multiple mechanisms [14,15].

To date, 18 HDAC isoforms have been found in humans and they can be divided into four classes. Classes I (1, 2, 3, 8), II (4, 5, 6, 7, 9, 10) and IV (11) HDACs are Zn^2+^-dependent enzymes, while class III HDACs (SIRT1-7) require NAD^+^ for their activities [16,17,18]. Five HDACis, vorinostat (1, SAHA) [19], bellinostat (2, PXD101) [20], panobinostat (3, LBH589) [21], romidepsin (4, FK228) [22] and chidamide (5, CS055) [23] (Figure 1), have been approved on the market for the treatment of cutaneous T-cell lymphoma (CTCL), multiple myeloma (MM) or peripheral T-cell lymphoma (PTCL), and more than 20 other inhibitors are at different stages of clinical trials. However, most HDACis displayed suboptimal results against solid tumors, so it is important to develop novel HDACis to achieve high potency against solid malignancy.

Despite the huge structural diversity, HDACis generally have a common pharmacophore model [24,25]: A zinc binding group (ZBG), a linker and a surface recognition group (CAP group) [26,27]. The CAP region is considered to be a key part for identifying different subtypes of HDACs, which interacts with the surface edge of the enzyme. The linker zone is negatively extended into the hydrophobic cavity of the HDACs [28]. HDACis are generally considered to be a group of promising candidates in anticancer drug discovery.

The coumarin derivatives have various pharmacological properties, such as antitumor, anti-inflammatory, antiviral and antimicrobial activities [29,30,31,32]. Previously, we designed and synthesized a series of coumarin-based hydroxamate HDACis, among which, compounds **6** and **7** displayed good inhibition against Class I HDACs [33]. Considering the comparable activities of compounds **6** and **7** and the synthetic convenience, we chose **7** as the initial lead to explore more coumarin-based HDACis with better activities (Figure 2). According to the preliminary docking results of **7** with HDAC1, it was found that the methoxyl group of the CAP region was in the cavity of the rim of the enzyme. Hence we wish to do some modifications on the methoxyl group for the sake of acquiring more valuable inhibitors. Details of the syntheses and biological evaluations of these new HDACis are presented below.

## 2. Chemistry

The syntheses of compounds **13a**–**g** and **14a**–**s** are shown in Scheme 1. Compound **8** was first subjected to a ring-closing reaction with diethyl carbonate and sodium hydride at 115 °C to give product **9**, which was subsequently reacted with methyl 8-bromooctanoate in *N*,*N*-dimethylformamid (DMF). The resulting ester was demethylated with BBr_3_ (1.0 M in dichloromethane (DCM)) to afford intermediate **11**, and **11** was further reacted with different alkyl bromides and then treated with hydroxylamine hydrochloride to obtain the target compounds **13a**–**g** and **14a**–**s**.

## 3. Results and Discussion

### 3.1. HDAC1 Inhibitory Activity of Synthesized Compounds

The obtained compounds were first tested for their inhibitory activities against HDAC1, and the FDA-approved marketed drug SAHA (**1**) was used as a positive control, and the obtained half maximal inhibitory concentration (IC_50_) values were shown in Table 1. In general, all synthesized samples were more active than the control drug (**1**), and the alkoxy-substituted coumarin derivatives (**13**-series) showed stronger inhibitory activity than most benzyloxy-substituted ones (**14**-series).

The **13**-series compounds (except **13g**) were 16- to 41-fold as active as SAHA (**1**) and they exhibited a linker-length-dependent inhibition toward HDAC1. The inhibitory activity of the target compounds improved with the elongation of the linker (**13a**–**c**), and **13c** showed the best activity with an IC_50_ of 0.30 nM. However, the inhibitory activity declined when the alkyl chain continued to extend (**13d**–**e**) or was replaced by a branched one (**13f**). Particularly, when the alkyl chain was linked with a cyclohexyl group (**13g**), a dramatic decrease of activity was observed. So the proper length and shape of the alkyl chain were very important to the HDAC1 inhibitory activity.

For the **14**-series compounds, the simplest **14a** showed an IC_50_ value of 0.96 nM, being 12 times more potent than SAHA (**1**). The inhibitory activities of these benzyloxy derivatives were significantly influenced by different substituents and substituting patterns on the benzyl ring, as analyzed below. Among the electron-withdrawing substituents on the mono-substituted benzyloxy fragment (**14b**–**l**), a trend of the inhibition was observed for fluoro > nitro > chloro > bromo > trifluoromethyl. When the fluorine was replaced by methyl group (**14p**–**r**), it resulted in a decrease of activity. At the same time, the efficacy of compounds was also obviously affected by the substituting position, and those with ortho-substitution (**14b**, **14e**, **14h** and **14p**) showed the best activity among the three investigated substituting sites (o-, m- and p-positions). Compound **14e** (IC_50_ = 0.75 nM) with an ortho-fluoro was the most potent inhibitor among all mono-substituted benzyloxy analogues, and the introduction of one more fluorine at the other ortho-position further improved the activity (**14m**, IC_50_ = 0.50 nM). However, the HDAC1 inhibitory activities of other disubstituted benzyloxy compounds (**14n** and **14o**) were not better than **14m**.

### 3.2. Antiproliferative Activity

According to the above-described enzyme inhibitory assay results, five of the most potent compounds (IC_50_ ≤ 0.50 nM Vs. 12.36 nM of the control drug SAHA) including four alkoxy-substituted derivatives (**13a**–**d**) and one benzyloxy-substituted analogue (**14m**) were further evaluated for their cellular level activities. The in vitro antiproliferative activities of these selected compounds against four human tumor cell lines MDA-MB-231, MCF-7, H157 and A549 were then tested using the SRB assay, and SAHA (**1**) was also used as the reference compound (Table 2). It was indicated that MDA-MB-231 cells were more sensitive to the tested compounds compared with other cancer cell lines. Notably, both **13a** (IC_50_ = 0.73 µM) and **13c** (IC_50_ = 0.36 µM) exhibited obviously better inhibitory activities than SAHA against all cell lines except A549, being 2~3-fold more potent than SAHA.

To assess whether the chosen compounds (**13a**–**d**) show selectivity between non-cancer cells and cancer cells, the following experiments were performed. Two normal cell lines were selected: human lung epithelial cells (Beas-2B) and human liver epithelial cells (L-02). As shown in Table 3, the results indicated that these compounds displayed no obvious cytotoxicity against the two human normal cells. Particularly, compound **13c** behaved even better than SAHA.

### 3.3. Colony Formation Assay

As all the tested compounds exhibited the best inhibitory activity against MDA-MB-231 cells, our subsequent work then focused on this tumor cell line. The antiproliferative activities of the two best compounds **13a** and **13c** were further verified by cell cloning experiment and SAHA (**1**) was used as the positive control. As depicted in Figure 3, when the concentrations of tested compounds were 0.25 μM, the effect was almost as good as that of the control drug at 0.5 μM. Both compounds resulted in a significant inhibition of the colony formation more potently than SAHA, and **13c** was especially active. The results suggested that compounds **13a** and **13c** could at least partly inhibit the growth and development of MDA-MB-231 cells.

### 3.4. Western Blot Analysis

Based on the aforementioned results, western blot analysis was further performed by immunoblotting assay (β-actin as the negative control). MDA-MB-231 cells were incubated with the vehicle alone, SAHA (5 µM), **13a** and **13c** (1, 2 and 5 µM) for 48 h, respectively. As shown in Figure 4, compounds **13a** and **13c** increased the acetylation levels of histone H3 and H4 in a clearly dose-dependent manner. Moreover, the results showed that the levels of acetyl-histone H3 and H4 in **13a** and **13c** treated groups were much higher than those in the control group, which was well consistent with their HDAC1 inhibitory activities.

### 3.5. Anti-Migration Assay

HDACis have been reported to suppress the migration of cancer cells [34]. We then evaluated the anti-metastatic activities of compounds **13a** and **13c** on MDA-MB-231 cells, and SAHA (**1**) was used as the positive control, as shown in Figure 5. In wound-healing assay, **13a** and **13c** could obviously inhibit the migration of MDA-MB-231 cells in a dose-dependent and time-dependent manner, both being more potent than SAHA. Even at 48 h, **13a** and **13c** still showed statistically significant anti-metastatic activity at the lowest testing concentration of 0.25 µM, compared with the SAHA-treated group at 1.0 µM (Figure 5B).

### 3.6. Cell Cycle Arrest Analysis

Next, we investigated whether the anti-proliferative activities of **13a** and **13c** resulted from the induction of cell cycle arrest. As shown in Figure 6, **13a** and **13c** arrested MDA-MB-231 cells in a dose-dependent manner. In comparison to the control group, compound **13c** arrested MDA-MB-231 cells mainly in G2/M phase (22.16% at 1 μM), which was obviously more potent than SAHA (11.02% at 1 μM).

### 3.7. Apoptosis Analysis

Finally, we determined whether the anti-proliferative effect of compounds **13a** and **13c** was related to the induction of cell apoptosis by flow cytometric analysis. As shown in Figure 7, MDA-MB-231 cells arrested were mainly in the early stage of apoptosis. **13a** and **13c** induced 13.24% and 19.26% cell apoptosis at 1 µM and induced 18.01% and 25.14% cell apoptosis at 2 µM, respectively. Meanwhile, SAHA (**1**) only induced 7.3% and 12.6% apoptotic cells at 1 and 2 µM, respectively. The quantitative analysis results clearly indicated that **13a** and **13c** were able to induce more apoptosis in MDA-MB-231 cells than the references compound **1**.

### 3.8. Molecular Docking Studies

Docking simulation was performed for the selected compound **13c** which was docked into HDAC1. Besides chelating with Zn^2+^, the hydroxamic acid group of **13c** formed one and two hydrogen bonds with His140 and His178, respectively (Figure 8). Moreover, π–π interaction was also observed between the phenyl of **13c** and Tyr204.

## 4. Experimental Section

### 4.1. Materials and Methods

Reagents were purchased from Energy Chemical Inc. (Shanghai, China) and Aladdin-reagents Inc. (Shanghai, China), and were used without further purification unless otherwise specified. All reactions were carried out with the use of standard techniques under an inert atmosphere. All target compounds were identified by ^1^H and ^13^C-NMR on a Bruker 600 MHz instrument (Bruker BioSpin AG, Fällanden, Switzerland), as well as HR-ESIMS data. High resolution mass spectra were gathered on an Agilent 6545 Q-TOF mass spectrometer (Agilent Technologies Inc., Waldbronn, Germany) operating in electrospray ionization (ESI) mode. Purity determination was conducted on an Agilent 1260 Series instrument (Agilent Technologies Inc., Waldbronn, Germany) using the following method: Eclipse XDB C18 column, 5 µm, 4.6 × 150 mm, column temperature 37 °C, 1.0 mL/min MeOH-H_2_O system, 60%–90% in 5 min, hold on 5 min, and then back to 60% in 5 min.

### 4.2. General Procedures for Synthesis of ***13a**–**g*** and ***14a**–**s***

Compound **8** (1.66 g, 10 mmol) was first subjected to a ring-closing reaction with diethyl carbonate (1.8 mL, 15 mmol) and sodium hydride (2.0 g, 50 mmol) in toluene (30 mL) at 115 °C to give product **9** (1.85 g, 60% yield), which was subsequently reacted with methyl 8-bromooctanoate (2.52 g, 10.59 mmol) in DMF. The compound **10** (2.31 g, 69% yield) was demethylated with BBr_3_ (1M in DCM) to afford intermediate **11** (1.99 g, 90% yield). Compound **11** (200 mg, 0.59 mmol) was further reacted with bromopropane (94.7 mg, 0.77 mmol).

To a solution of NH_2_OH·HCl (875 mg, 12.6 mmol) in 8 mL MeOH, KOH (706 mg, 12.6 mmol) was added. Then the reaction mixture was stirred at 40 °C for 10 min and cooled to 0 °C and filtered. Methyl-8-((2-oxo-7-propoxy-2*H*-chromen-4-yl)oxy) octanoate (137 mg, 63% yield) was added to the filtrate followed by KOH (76.3 mg, 1.36 mmol), after which the reaction was stirred at room temperature for 30 min. The solvent was removed and extracted with EtOAc. The obtained residue was finally purified by column chromatography [CH_2_Cl_2_:MeOH = 10:1] to give compound **13a**. Compounds **13b**–**g** and **14a**–**s** were synthesized according to the procedure described for the preparation of **13a**.

*N-Hydroxy-8-((2-oxo-7-propoxy-2H-benzopyran-4-yl)oxy)octanoic acid* (**13a**). Yield: 33%. ^1^H-NMR (600 MHz, CD_3_OD) δ 7.74 (d, *J* = 8.4 Hz, 1H), 6.94–6.86 (m, 2H), 5.64 (s, 1H), 5.49 (s, 2H), 4.20 (t, *J* = 6.6 Hz, 2H), 4.03 (t, *J* = 6.6 Hz, 2H), 2.10 (t, *J* = 7.2 Hz, 2H), 1.93–1.88 (m, 2H), 1.67–1.52 (m, 4H), 1.47–1.36 (m, 4H), 1.06 (t, *J* = 7.2 Hz, 3H). ^13^C-NMR (150 MHz, CD_3_OD) δ 172.98, 168.36, 166.09, 164.61, 156.40, 125.24, 113.96, 109.97, 101.91, 88.23, 71.29, 70.82, 33.72, 30.00, 29.97, 29.54, 26.89, 26.65, 23.45, 10.75. HRMS (ESI) calcd for C_20_H_27_NO_6_ [M + H]^+^, 378.1911; found 378.1909. HPLC purity: 98.55%, t_R_ = 6.633 min.

*N-Hydroxy-8-((7-butoxy-2-oxo-2H-benzopyran-4-yl)oxy)octanoic acid* (**13b**). Yield: 26%. ^1^H-NMR (600 MHz, DMSO-*d_6_*) δ 10.34 (brs, 1H), 8.67 (brs, 1H), 7.67 (d, *J* = 7.8 Hz, 1H), 6.95–6.93 (m, 2H), 5.71 (s, 1H), 4.17 (t, *J* = 6.6 Hz, 2H), 4.06 (t, *J* = 6.6 Hz, 2H), 1.95 (t, *J* = 7.2 Hz, 2H), 1.81–1.69 (m, 4H), 1.52–1.42 (m, 6H), 1.36–1.24 (m, 4H), 0.94 (t, *J* = 7. 2 Hz, 3H). ^13^C-NMR (150 MHz, DMSO-*d_6_*) δ 169.10, 165.45, 162.33, 162.15, 154.64, 123.89, 112.54, 108.26, 100.94, 87.79, 69.27, 67.99, 32.24, 30.49, 28.48, 28.36, 27.94, 25.31, 25.06, 18.68, 13.67. HRMS (ESI) calcd for C_21_H_29_NO_6_ [M + H]^+^ 392.2068; found 392.2067. HPLC purity: 98.70%, t_R_ = 7.665 min.

*N-Hydroxy-8-((7-(2-methoxyethoxy)-2-oxo-2H-chromen-4**-yl)oxy)octanoic acid* (**13c**). Yield: 27%. ^1^H-NMR (600 MHz, CD_3_OD) δ 7.76 (d, *J* = 7.8 Hz, 1H), 6.97–6.91 (m, 2H), 5.66 (s, 1H), 4.22–4.19 (m, 4H), 3.78–3.77 (m, 2H), 3.43 (s, 3H), 2.10 (t, *J* = 7.2 Hz, 2H), 1.93–1.89 (m, 2H), 1.67–1.53 (m, 4H), 1.47–1.37 (m, 4H). ^13^C-NMR (150 MHz, CD_3_OD) δ 168.29, 166.00, 164.25, 156.35, 125.32, 113.97, 110.29, 102.13, 88.41, 71.90, 70.85, 69.12, 59.27, 33.71, 30.00, 29.97, 29.54, 26.89, 26.65. HRMS (ESI) calcd for C_20_H_27_NO_7_ [M + H]^+^ 394.1860; found 394.1863. HPLC purity: 95.43%, t_R_ = 8.571 min.

*N-Hydroxy-8-((7-(3-methoxypropoxy)-2-oxo-2H-chromen-4**-yl)oxy)octanoic acid* (**13d**). Yield: 23%. ^1^H-NMR (600 MHz, DMSO-*d_6_*) δ 10.32 (brs, 1H), 8.65 (brs, 1H), 7.68 (d, *J* = 7.8 Hz, 1H), 6.95–6.94 (m, 2H), 5.75 (s, 1H), 4.17 (t, *J* = 6.6 Hz, 2H), 4.11 (t, *J* = 6.6 Hz, 2H), 3.47 (t, *J* = 6.6 Hz, 2H), 3.25 (s, 3H), 1.99–1.93 (m, 4H), 1.82–1.79 (m, 2H), 1.52–1.41 (m, 4H), 1.36–1.24 (m, 4H). ^13^C-NMR (150 MHz, DMSO-*d_6_*) δ 169.11, 165.42, 162.18, 162.13, 154.61, 123.94, 112.47, 108.36, 101.02, 87.86, 69.28, 68.30, 65.42, 57.98, 32.24, 28.72, 28.47, 28.35, 27.93, 25.30, 25.05. HRMS (ESI) calcd for C_21_H_29_NO_7_ [M + H]^+^ 408.2017; found 408.2015. HPLC purity: 98.36%, t_R_ = 5.657 min.

*N-Hydroxy-8-((7-(2-(benzyloxy)ethoxy)-2-oxo-2H-chromen-4**-yl)oxy)octanamide* (**13e**). Yield: 29%. ^1^H-NMR (600 MHz, DMSO-*d_6_*) δ 10.33 (brs, 1H), 8.65 (brs, 1H), 7.67 (d, *J* = 8.4 Hz, 1H), 7.35–7.27 (m, 5H), 7.00–6.96 (m, 2H), 5.72 (s, 1H), 4.56 (s, 2H), 4.26–4.24 (m, 2H), 4.18 (t, *J* = 6.0 Hz, 2H), 3.80–3.78 (m, 2H), 1.94 (t, *J* = 7.2 Hz, 2H), 1.82–1.77 (m, 2H), 1.52–1.41 (m, 4H), 1.36–1.24 (m, 4H). ^13^C-NMR (150 MHz, DMSO-*d_6_*) δ 169.09, 165.40, 162.10, 154.57, 138.22, 128.26, 127.56, 127.48, 123.92, 112.61, 108.42, 101.08, 87.89, 72.07, 69.27, 67.94, 67.87, 32.23, 28.46, 28.34, 27.92, 25.29, 25.04. HRMS (ESI) calcd for C_26_H_31_NO_7_ [M + H]^+^ 470.2173; found 470.2170. HPLC purity: 97.06%, t_R_ = 8.193 min.

*N-Hydroxy-8-((7-isobutoxy-2-oxo-2H-chromen-4-yl)oxy)octanoic acid* (**13f**). Yield: 27%. ^1^H-NMR (600 MHz, CD_3_OD) δ 7.74 (d, *J* = 8.4 Hz, 1H), 6.94–6.85 (m, 2H), 5.64 (s, 1H), 4.19 (t, *J* = 6.6 Hz, 2H), 3.84 (d, *J* = 6.6 Hz, 2H), 2.11–2.09 (m, 3H), 1.92–1.88 (m, 2H), 1.67–1.52 (m, 4H), 1.47–1.39 (m, 4H), 1.05 (d, *J* = 6.6 Hz, 6H). ^13^C-NMR (150 MHz, CD_3_OD) δ 172.96, 168.35, 166.08, 164.66, 156.39, 125.24, 113.95, 109.98, 101.92, 88.24, 76.04, 70.82, 33.72, 30.00, 29.98, 29.54, 29.42, 26.89, 26.65, 19.42. HRMS (ESI) calcd for C_21_H_29_NO_6_ [M + H]^+^ 392.2068; found 392.2071. HPLC purity: 98.45%, t_R_ = 7.239 min.

*N-Hydroxy-8-((7-(cyclohexyl)-2-oxo-2H-benzopyran-4-yl)oxy)octanoic acid* (**13g**). Yield: 30%. ^1^H-NMR (600 MHz, DMSO-*d_6_*) δ 10.33 (brs, 1H), 8.65 (brs, 1H), 7.67 (d, *J* = 7.8 Hz, 1H), 6.95–6.93 (m, 2H), 5.71 (s, 1H), 4.17 (t, *J* = 6.0 Hz, 2H), 3.88 (d, *J* = 5.4 Hz, 2H), 1.94 (t, *J* = 7.2 Hz, 2H), 1.81–1.64 (m, 8H), 1.52–1.43 (m, 4H), 1.36–1.23 (m, 7H), 1.38–1.02 (m, 2H). ^13^C-NMR (150 MHz, DMSO-*d_6_*) δ 169.10, 165.44, 162.43, 162.14, 154.62, 123.89, 112.53, 108.23, 100.98, 87.79, 73.34, 69.26, 36.90, 32.23, 29.12, 28.46, 28.34, 27.93, 25.99, 25.29, 25.21, 25.04. HRMS (ESI) calcd for C_24_H_33_NO_6_ [M + H]^+^ 432.2381; found 432.2383. HPLC purity: 97.11%, t_R_ = 8.778 min.

*N-Hydroxy-8-((7-(oxy)-2-oxo-2H-benzopyran-4**-yl)oxy)octanoic acid* (**14a**). Yield: 31%. ^1^H-NMR (600 MHz, DMSO-*d**_6_*) δ 10.33 (brs, 1H), 8.65 (brs, 1H), 7.70 (d, *J* = 7.8 Hz, 1H), 7.47 (d, *J* = 7.8 Hz, 2H), 7.41 (dd, *J* = 7.8 Hz, 7.8 Hz, 2H), 7.35 (dd, *J* = 7.8 Hz, 7.8 Hz, 1H), 7.07–7.02 (m, 2H), 5.73 (s, 1H), 5.21 (s, 2H), 4.18 (t, *J* = 6.6 Hz, 2H), 1.94 (t, *J* = 7.2 Hz, 2H), 1.82–1.77 (m, 2H), 1.52–1.41 (m, 4H), 1.36–1.23 (m, 4H). ^13^C-NMR (150 MHz, DMSO-*d_6_*) δ 169.10, 165.38, 162.08, 161.88, 154.54, 136.22, 128.53, 128.13, 127.95, 123.97, 112.83, 108.57, 101.47, 87.96, 69.90, 69.29, 32.23, 28.47, 28.34, 27.93, 25.29, 25.04. HRMS (ESI) calcd for C_24_H_27_NO_6_ [M + H]^+^ 426.1911; found 426.1906. HPLC purity: 96.83%, t_R_ = 8.498 min.

*N-Hydroxy-8-((7-((2-nitrobenzyl)oxy)-2-oxo-2H-chromen-4**-yl)oxy)octanoic acid* (**14b**). Yield: 23%. ^1^H-NMR (600 MHz, DMSO- *d**_6_*) δ 10.33 (brs, 1H), 8.65 (brs, 1H), 7.80–7.69 (m, 2H), 7.53–7.35 (m, 3H), 7.09–7.01 (m, 2H), 5.75–5.73 (m, 1H), 5.21–5.12 (m, 2H), 4.18 (s, 2H), 1.94 (t, *J* = 7.2 Hz, 2H), 1.80 (s, 2H), 1.52–1.43 (m, 4H), 1.35–1.23 (m, 4H). ^13^C-NMR (150 MHz, DMSO-*d_6_*) δ 169.11, 165.37, 162.09, 154.54, 130.42, 130.18, 128.54, 126.81, 125.09, 118.93, 112.61, 108.68, 101.41, 69.31, 65.78, 32.23, 28.46, 28.34, 27.93, 25.29, 25.04. HRMS (ESI) calcd for C_24_H_26_N_2_O_8_ [M + H]^+^ 471.1762; found 471.1761. HPLC purity: 98.92%, t_R_ = 8.400 min.

*N-Hydroxy-8-((7-((3-nitrobenzyl)oxy)-2-oxo-2H-chromen-4**-yl)oxy)octanoic acid* (**14c**). Yield: 22%. ^1^H-NMR (600 MHz, CD_3_OD) δ 7.75 (d, *J* = 8.6 Hz, 1H), 7.46 (d, *J* = 7.2 Hz, 1H), 7.39 (dd, *J* = 7.2 Hz, 7.2 Hz, H), 7.33 (dd, *J* = 7.2 Hz, 7.2 Hz, 1H), 7.16–7.00 (m, 2H), 6.95 (s, 1H), 5.49 (s, 1H), 5.17 (s, 2H), 4.19 (t, *J* = 6.0 Hz, 2H), 2.10 (t, *J* = 7.2 Hz, 2H), 1.91–1.89 (m, 2H), 1.67–1.53 (m, 4H), 1.44–1.38 (m, 4H). ^13^C-NMR (150 MHz, CD_3_OD) δ 172.98, 168.25, 165.98, 164.08, 156.30, 137.71, 129.64, 129.22, 128.75, 125.31, 119.67, 119.00, 114.25, 110.31, 102.54, 88.42, 71.58, 70.84, 33.72, 29.98, 29.97, 29.53, 26.88, 26.64. HRMS (ESI) calcd for C_24_H_26_N_2_O_8_ [M + H]^+^ 471.1762; found 471.1761. HPLC purity: 98.94%, t_R_ = 8.391 min.

*N-Hydroxy-8-((7-((4-nitrobenzyl)oxy)2-oxo-2H-chromen-4-yl)oxy)octanoic acid* (**14d**). Yield: 23%. ^1^H-NMR (600 MHz, DMSO-*d_6_*) δ 10.33 (brs, 1H), 8.65 (brs, 1H), 8.27 (d, *J* = 7.8 Hz, 1H), 7.75–7.68 (m, 2H), 7.58–7.35 (m, 2H), 7.16–7.06 (m, 2H), 5.73 (d, *J* = 8.4 Hz, 1H), 5.40–5.20 (m, 2H), 4.18 (t, *J* = 6.0 Hz, 2H), 1.94 (t, *J* = 7.2 Hz, 2H), 1.81–1.77 (m, 2H), 1.52–1.41 (m, 4H), 1.36–1.24 (m, 4H). ^13^C-NMR (150 MHz, DMSO-*d_6_*) δ 169.10, 165.35, 162.06, 161.67, 154.52, 135.29, 132.71, 129.76, 128.54, 124.00, 112.81, 108.68, 101.51, 88.02, 69.31, 69.01, 32.23, 28.46, 28.34, 27.92, 25.29, 25.04. HRMS (ESI) calcd for C_24_H_26_N_2_O_8_ [M + H]^+^ 471.1758; found 471.1762. HPLC purity: 99.00%, t_R_ = 8.393 min.

*N-Hydroxy-8-((7-((2-fluorobenzyl)oxy)-2-oxo-2H-chromen**-4-yl)oxy)octanoic acid* (**14e**). Yield: 28%. ^1^H-NMR (600 MHz, DMSO-*d_6_*) δ 10.33 (brs, 1H), 8.65 (brs, 1H), 7.71 (d, *J* = 8.4 Hz, 1H), 7.60–7.58 (dd, *J* = 7.8 Hz, 7.8 Hz, 1H), 7.47–7.44 (m, 1H), 7.29–7.24 (m, 2H), 7.12–7.02 (m, 2H), 5.74 (s, 1H), 5.25 (s, 2H), 4.18 (t, *J* = 6.0 Hz, 2H), 1.94 (t, *J* = 7.2 Hz, 2H), 1.82–1.77 (m, 2H), 1.52–1.42 (m, 4H), 1.36–1.23 (m, 4H). ^13^C-NMR (150 MHz, DMSO-*d_6_*) δ 169.13, 165.38, 162.11, 161.71, 154.56, 131.05, 130.86, 124.64, 124.05, 123.08, 115.59, 115.46, 112.70, 18.76, 101.43, 88.07, 69.33, 64.32, 32.25, 28.48, 28.36, 27.94, 25.31, 25.06. HRMS (ESI) calcd for C_24_H_26_FNO_6_ [M + H]^+^ 444.1817; found 444.1815. HPLC purity: 96.93%, t_R_ = 8.151 min.

*N-Hydroxy-8-((7-((3-fluorobenzyl)oxy)-2-oxo-2H-chromen**-4-yl)oxy)octanoic acid* (**14f**). Yield: 29%. ^1^H-NMR (600 MHz, DMSO-*d_6_*) δ 10.32 (brs, 1H), 8.65 (brs, 1H), 7.70 (d, *J* = 7.8 Hz, 1H), 7.47–7.44 (m, 1H), 7.32 (d, *J* = 7.2 Hz, 2H), 7.20–7.17 (m, 1H), 7.07–7.03 (m, 2H), 5.73 (s, 1H), 5.24 (s, 2H), 4.18 (t, *J* = 6.3 Hz, 2H), 1.94 (t, *J* = 7.2 Hz, 2H), 1.82–1.77 (m, 2H), 1.52–1.41 (m, 4H), 1.36–1.24 (m, 4H). ^13^C-NMR (150 MHz, DMSO-*d_6_*) δ 169.09, 165.34, 162.05, 161.63, 154.51, 139.12, 130.62, 124.01, 123.78, 123.76, 114.94, 114.54, 112.79, 108.72, 101.52, 88.04, 69.30, 56.02, 32.23, 30.70, 28.46, 27.92, 25.29, 25.04. HRMS (ESI) calcd for C_24_H_26_FNO_6_ [M + H]^+^ 444.1817; found 444.1816. HPLC purity: 97.07%, t_R_ = 8.510 min.

*N-Hydroxy-8-((7-((4-fluorobenzyl)oxy)2-oxo-2H-chromen-4yl)oxy)octanoic acid* (**14g**). Yield: 30%. ^1^H-NMR (600 MHz, DMSO-*d_6_*) δ 10.33 (brs, 1H), 8.66 (brs, 1H), 7.70 (d, *J* = 8.8 Hz, 1H), 7.54–7.52 (m, 2H), 7.25–7.22 (m, 2H), 7.07–7.01 (m, 2H), 5.73 (s, 1H), 5.19 (s, 2H), 4.18 (t, *J* = 6.3 Hz, 2H), 1.94 (t, *J* = 7.2 Hz, 2H), 1.83–1.78 (m, 2H), 1.52–1.41 (m, 4H), 1.36–1.24 (m, 4H). ^13^C-NMR (150 MHz, DMSO-*d_6_*) δ 169.13, 165.39, 162.11, 161.79, 154.55, 132.47, 130.34, 124.00, 115.46, 115.31, 112.84, 108.63, 101.49, 88.00, 69.32, 69.18, 32.25, 28.48, 28.35, 27.94, 25.30, 25.06. HRMS (ESI) calcd for C_24_H_26_FNO_6_ [M + H]^+^ 444.1817; found 444.1819. HPLC purity: 98.68%, t_R_ = 8.446 min.

*N-Hydroxy-8-((7-((2-chlorobenzyl)oxy)-2-oxo-2H-chromen**-4-yl)oxy)octanoic acid* (**14h**). Yield: 23%. ^1^H-NMR (600 MHz, DMSO-*d_6_*) δ 10.33 (brs, 1H), 8.65 (brs, 1H), 7.71 (d, *J* =8.4 Hz, 1H), 7.63 (d, *J* = 7.2 Hz, 1H), 7.54 (d, *J* = 7.2 Hz, 1H), 7.44–7.39 (m, 2H), 7.11–7.03 (m, 2H), 5.74 (s, 1H), 5.26 (s, 2H), 4.18 (t, *J* = 6.0 Hz, 2H), 1.94 (t, *J* = 7.2 Hz, 2H), 1.83–1.77 (m, 2H), 1.52–1.42 (m, 4H), 1.36–1.24 (m, 4H). ^13^C-NMR (150 MHz, DMSO-*d_6_*) δ 169.10, 165.34, 162.06, 161.70, 154.53, 133.51, 132.96, 130.61, 130.27, 129.51, 127.47, 124.06, 112.63, 108.81, 101.48, 88.07, 69.31, 67.61, 32.23, 28.47, 28.34, 27.92, 25.29, 25.04. HRMS (ESI) calcd for C_24_H_26_ClNO_6_ [M + H]^+^ 460.1521; found 460.1522. HPLC purity: 98.15%, t_R_ = 9.287 min.

*N-Hydroxy-8-((7-((3-chlorobenzyl)oxy)-2-oxo-2H-chromen-4-yl)oxy)octanoic acid* (**14i**). Yield: 33%. ^1^H-NMR (600 MHz, DMSO-*d_6_*) δ 10.33 (brs, 1H), 8.65 (brs, 1H), 7.70 (d, *J* = 7.8 Hz, 1H), 7.55 (s, 1H), 7.45–7.42 (m, 3H), 7.07–7.03 (m, 2H), 5.73 (s, 1H), 5.23 (s, 2H), 4.18 (t, *J* = 6.6 Hz, 2H), 1.94 (t, *J* = 7.2 Hz, 2H), 1.82–1.77 (m, 2H), 1.52–1.41 (m, 4H), 1.36–1.23 (m, 4H). ^13^C-NMR (150 MHz, DMSO-*d_6_*) δ 169.11, 165.35, 162.06, 161.61, 154.52, 138.82, 133.18, 130.48, 128.04, 127.55, 126.43, 124.04, 112.79, 108.74, 101.52, 88.05, 69.32, 67.19, 32.24, 28.47, 28.35, 27.93, 25.29, 25.05. HRMS (ESI) calcd for C_24_H_26_ClNO_6_ [M + H]^+^ 460.1521; found 460.1522. HPLC purity: 98.97%, t_R_ = 9.096 min.

*N-Hydroxy-8-((7-((4-chlorobenzyl)oxy)-2-oxo-2H-chromen**-4-yl)oxy)octanoic acid* (**14j**). Yield: 31%. ^1^H-NMR (600 MHz, DMSO-*d_6_*) δ 10.33 (brs, 1H), 8.65 (brs, 1H), 7.70 (d, *J* = 8.4 Hz, 1H), 7.51 (d, *J* = 8.4 Hz, 2H), 7.47 (d, *J* = 8.4 Hz, 2H), 7.06–7.01 (m, 2H), 5.73 (s, 1H), 5.22 (s, 2H), 4.18 (t, *J* = 6.6 Hz, 2H), 1.94 (t, *J* = 7.2 Hz, 2H), 1.81–1.77 (m, 2H), 1.52–1.41 (m, 4H), 1.36–1.23 (m, 4H). ^13^C-NMR (150 MHz, DMSO-*d_6_*) δ 169.10, 165.35, 162.06, 161.67, 154.52, 135.29, 132.71, 129.76, 128.54, 124.00, 112.81, 108.68, 101.51, 88.02, 69.31, 69.01, 32.23, 28.47, 28.35, 27.93, 25.29, 25.05. HRMS (ESI) calcd for C_24_H_26_ClNO_6_ [M + H]^+^460.1521; found 460.1518. HPLC purity: 97.14%, t_R_ = 6.715 min.

*N-Hydroxy-8-((7-((4-bromobenzyl)oxy)2-oxo-2H-chromen-4**-yl)oxy)octanoic acid* (**14k**). Yield: 32%. ^1^H-NMR (600 MHz, DMSO-*d_6_*) δ 10.33 (brs, 1H), 8.65 (brs, 1H), 7.70 (d, *J* = 8.4 Hz, 1H), 7.61 (d, *J* = 8.4 Hz, 2H), 7.43 (d, *J* = 7.8 Hz, 2H), 7.06–7.01 (m, 2H), 5.73 (s, 1H), 5.20 (s, 2H), 4.18 (t, *J* = 6.0 Hz, 2H), 1.94 (t, *J* = 7.2 Hz, 2H), 1.81–1.77 (m, 2H), 1.52–1.41 (m, 4H), 1.36–1.23 (m, 4H). ^13^C-NMR (150 MHz, DMSO-*d_6_*) δ 169.11, 165.36, 162.07, 161.66, 154.52, 135.72, 131.47, 130.05, 124.01, 121.27, 112.83, 108.69, 101.52, 88.03, 69.31, 69.04, 32.24, 28.47, 28.34, 27.92, 25.29, 25.04. HRMS (ESI) calcd for C_24_H_26_BrNO_6_ [M + H]^+^ 504.1016; found 504.1018. HPLC purity: 96.29%, t_R_ = 9.300 min.

*N-Hydroxy-8-((7-((4-trifluoromethyl)oxy))-2-oxo-2H-chromen-4-yl)oxy)octanoic acid* (**14l**). Yield: 31%. ^1^H-NMR (600 MHz, DMSO-*d_6_*) δ 10.33 (brs, 1H), 8.66 (brs, 1H), 7.78 (d, *J* = 7.8 Hz, 2H), 7.71–7.69 (m, 3H), 7.08–7.04 (m, 2H), 5.73 (s, 1H), 5.34 (s, 2H), 4.17 (t, *J* = 6.0 Hz, 2H), 1.94 (t, *J* = 7.8 Hz, 2H), 1.80–1.78 (m, 2H), 1.52–1.41 (m, 4H), 1.36–1.22 (m, 4H). ^13^C-NMR (150 MHz, DMSO-*d_6_*) δ 169.11, 165.33, 162.04, 161.55, 154.52, 141.14, 128.23, 125.52, 125.44, 125.42, 125.39, 125.11, 124.07, 112.78, 108.80, 101.55, 88.07, 69.32, 68.92, 32.24, 28.47, 28.35, 27.93, 25.29, 25.05. HRMS (ESI) calcd for C_24_H_26_F_3_NO_6_ [M + H]^+^ 494.1785; found 494.1784. HPLC purity: 95.43%, t_R_ = 8.818 min.

*N-Hydroxy-8-((7-((2,6-difluorobenzyl)oxy)2-oxo-2H-chromen-4-yl)oxy)octanoic acid* (**14m**). Yield: 29%. ^1^H-NMR (600 MHz, DMSO-*d_6_*) δ 10.33 (brs, 1H), 8.65 (brs, 1H), 7.71 (d, *J* = 7.8 Hz, 1H), 7.58–7.53 (m, 1H), 7.22–7.15 (m, 3H), 7.03–7.01 (m, H), 5.75 (s, 1H), 5.23 (s, 2H), 4.18 (t, *J* = 6.0 Hz, 2H), 1.94 (t, *J* = 7.2 Hz, 2H), 1.82–1.77 (m, 2H), 1.52–1.42 (m, 4H), 1.36–1.24 (m, 4H). ^13^C-NMR (150 MHz, DMSO-*d_6_*) δ 169.12, 165.33, 162.07, 162.01, 161.54, 160.40, 154.54, 132.13, 124.07, 112.58, 112.00, 111.83, 111.54, 108.94, 101.33, 88.16, 69.34, 58.30, 32.24, 28.48, 28.36, 27.93, 25.30, 25.06. HRMS (ESI) calcd for C_24_H_25_F_2_NO_6_ [M + H]^+^ 462.1723; found 462.1719. HPLC purity: 97.95%, t_R_ = 8.157 min.

*N-Hydroxy-8-((7-((2,4-difluorobenzyl)oxy)2-oxo-2H**-chromen-4-yl)oxy)octanoic acid* (**14n**). Yield: 28%. ^1^H-NMR (600 MHz, DMSO-*d_6_*) δ 10.33 (brs, 1H), 8.65 (brs, 1H), 7.71–7.65 (m, 2H), 7.35–7.31 (m, 1H), 7.16–7.12 (m, 2H), 7.03–7.02 (m, H), 5.74 (s, 1H), 5.21 (s, 2H), 4.18 (t, *J* = 6.0 Hz, 2H), 1.94 (t, *J* = 7.2 Hz, 2H), 1.82–1.77 (m, 2H) 1.52–1.42 (m, 4H), 1.36–1.24 (m, 4H). ^13^C-NMR (150 MHz, DMSO-*d_6_*) δ 169.10, 165.34, 162.06, 161.59, 154.53, 132.52, 124.03, 119.60, 119.49, 112.66, 117.71, 108.78, 104.28, 104.08, 101.36, 88.09, 69.32, 63.85, 32.23, 28.46, 28.34, 27.92, 25.29, 25.04. HRMS (ESI) calcd for C_24_H_25_F_2_NO_6_ [M + H]^+^ 462.1723; found 462.1726. HPLC purity: 97.94%, t_R_ = 7.569 min.

*N-Hydroxy-8-((7-((2-chloro-4-fluorobenzyl)oxy)2-oxo-2H-chromen-4-yl)oxy)octanoic acid* (**14o**). Yield: 30%. ^1^H-NMR (600 MHz, DMSO-*d_6_*) δ 10.32 (brs, 1H), 8.65 (brs, 1H), 7.72–7.69 (m, 2H), 7.56–7.54 (m, H), 7.31–7.28 (m, H), 7.13–7.03 (m, 2H), 5.75 (s, 1H), 5.24 (s, 2H), 4.19 (t, *J* = 6.6 Hz, 2H), 1.94 (t, *J* = 7.2 Hz, 2H), 1.82–1.78 (m, 2H), 1.52–1.42 (m, 4H), 1.37–1.25 (m, 4H). ^13^C-NMR (150 MHz, DMSO-*d_6_*) δ 169.10, 165.34, 162.06, 161.64, 154.53, 132.47, 130.06, 124.06, 117.06, 116.89, 114.69, 114.55, 112.65, 108.85, 101.49, 88.10, 69.32, 67.08, 32.23, 28.46, 28.34, 27.92, 25.29, 25.04. HRMS (ESI) calcd for C_24_H_25_F_2_NO_6_ [M + H]^+^ 478.1427; found 478.1432. HPLC purity: 95.38%, t_R_ = 7.474 min.

*N-Hydroxy-8-((7-((2-methylbenzyl)oxy)-2-oxo-2H-chromen**-4-yl)oxy)octanoic acid* (**14p**). Yield: 29%. ^1^H-NMR (600 MHz, DMSO-*d_6_*) δ 10.33 (brs, 1H), 8.66 (brs, 1H), 7.71 (d, *J* = 7.2 Hz, 1H), 7.43 (d, *J* = 7.2 Hz, 1H), 7.27–7.20 (m, 3H), 7.11–7.03 (m, 2H), 5.73 (s, 1H), 5.20 (s, 2H), 4.18 (t, *J* = 6.6 Hz, 2H), 2.33 (s, 3H), 1.94 (t, *J* = 7.2 Hz, 2H), 1.82–1.77 (m, 2H), 1.52 –1.41 (m, 4H), 1.36–1.24 (m, 4H). ^13^C-NMR (150 MHz, DMSO-*d_6_*) δ 169.12, 165.41, 162.11, 162.01, 154.58, 136.80, 134.16, 130.23, 128.74, 128.36, 125.86, 123.97, 112.78, 108.59, 101.44, 87.95, 69.30, 68.66, 32.24, 28.48, 28.35, 27.93, 25.30, 25.05, 18.48. HRMS (ESI) calcd for C_25_H_29_NO_6_ [M + H]^+^ 440.2068; found 440.2072. HPLC purity: 93.79%, t_R_ = 6.635 min.

*N-Hydroxy-8-((7-((3-methylbenzyl)oxy)-2-oxo-2H-chromen-4-yl)oxy)octanoic acid* (**14q**). Yield: 31%. ^1^H-NMR (600 MHz, CD_3_OD) δ 7.73 (d, *J* = 8.4 Hz, 1H), 7.27–7.22 (m, 3H), 7.15 (d, *J* = 7.2 Hz, 1H), 6.99–6.92 (m, 2H), 5.63 (s, 1H), 5.12 (s, 2H), 4.17 (s, 2H), 2.35 (s, 3H), 2.10 (t, *J* = 7.2 Hz, 2H), 1.89 (s, 2H), 1.65–1.53 (m, 4H), 1.43–1.38 (m, 4H). ^13^C-NMR (150 MHz, CD_3_OD) δ 172.97, 168.23, 165.97, 164.10, 156.27, 139.47, 137.61, 129.88, 129.54, 129.33, 125.81, 125.27, 114.24, 110.25, 102.50, 88.40, 71.61, 70.83, 33.72, 29.99, 29.97, 29.53, 26.88, 26.64, 21.43. HRMS (ESI) calcd for C_25_H_29_NO_6_ [M + H]^+^ 440.2068; found 440.2070. HPLC purity: 97.60%, t_R_ = 6.704 min.

*N-Hydroxy-8-((7-((4-methylbenzyl)oxy)2-oxo-2H-chromen**-4-yl)oxy)octanoic acid* (**14r**). Yield: 33%. ^1^H-NMR (600 MHz, DMSO-*d_6_*) δ 10.33 (brs, 1H), 8.65 (brs, 1H), 7.68 (d, *J* = 8.4 Hz, 1H), 7.36 (d, *J* = 7.8 Hz, 2H), 7.21 (d, *J* = 7.8 Hz, 2H), 7.04–6.99 (m, 2H), 5.72 (s, 1H), 5.16 (s, 2H), 4.17 (t, *J* = 6.0 Hz, 2H), 2.31 (s, 3H), 1.94 (t, *J* = 7.2 Hz, 2H), 1.81–1.77 (m, 2H), 1.52–1.41 (m, 4H), 1.36–1.24 (m, 4H). ^13^C-NMR (150 MHz, DMSO-*d_6_*) δ 169.09, 165.38, 162.08, 161.90, 154.53, 137.41, 133.17, 129.06, 128.04, 123.92, 112.85, 108.49, 101.44, 87.92, 69.81, 69.28, 32.23, 28.46, 28.34, 27.92, 25.29, 25.04, 20.79. HRMS (ESI) calcd for C_25_H_29_NO_6_ [M + H]^+^ 440.2068; found 440.2072. HPLC purity: 97.62%, t_R_ = 9.009 min.

*N-Hydroxy-8-((7-((2,6-dimethylbenzyl)oxy)-2-oxo-2H-chromen-4-yl)oxy)octanoic acid* (**14s**). Yield: 29%. ^1^H-NMR (600 MHz, CD_3_OD) δ 7.79 (d, *J* = 8.4 Hz, 1H), 7.15 (dd, *J* = 7.2 Hz, 7.2 Hz, 1H), 7.08–7.06 (m, 3H), 7.02–7.00 (m, H), 5.67 (s, 1H), 5.19 (s, 2H), 4.21 (t, *J* = 6.0 Hz, 2H), 2.37 (s, 6H), 2.10 (t, *J* = 7.2 Hz, 2H), 1.94–1.89 (m, 2H), 1.67–1.53 (m, 4H), 1.47–1.37 (m, 4H). ^13^C-NMR (150 MHz, CD_3_OD) δ 172.97, 168.32, 166.04, 164.64, 156.43, 139.33, 133.45, 129.83, 129.30, 125.35, 114.10, 110.31, 102.15, 88.41, 70.86, 66.58, 32.72, 30.00, 29.98, 29.55, 26.90, 26.65, 19.60. HRMS (ESI) calcd for C_26_H_31_NO_6_ [M + H]^+^ 454.2224; found 454.2221. HPLC purity: 97.43%, t_R_ = 7.566 min.

Spectra of ^1^H and ^13^C-NMR and HR MS and HPLC for new compounds **13a**–**g** and **14a**–**s** can be found in Appendix A.

### 4.3. HDAC1 Inhibitory Assay

The enzyme inhibitory assay was conducted by Shanghai ChemPartner Corporation (Shanghai, China). Briefly, HDAC1 enzyme solution was incubated with SAHA or target compounds at different concentrations in the presence of HDAC substrate (Boc-Lys(Ac)-AMC) at 37 °C for 60 min. Then the lysine developer was added to terminate the reaction, and the samples were further incubated at 37 °C for 30 min. The data were recorded on an ELISA plate reader at 405 nm. Three independent experiments with triplicate were carried out.

### 4.4. Cell Culture and Reagents

The cells used in the experiment included two human breast cancer cell lines (MCF-7, MDA-MB-231) and two human lung adenocarcinoma cell lines (H157, A549). Cancer cells (6 × 10^3^ per well) were inoculated into 96-well plates. The treatment was carried out for 48 h, followed by SRB staining, and the absorption value was measured at 540 nm by a microplate reader. SAHA was used as a positive control, and the independent experiment was performed three times.

### 4.5. Colony Formation Assay

Six-well plates were inoculated with MDA-MB-231 cells (800/well). The plate was continuously cultured for four days until the cells became agglomerated, and then treated with SAHA and **13a**, **13c**, change the dosing medium every 24h. One week after the addition of the drugs, the medium was removed. The cells were then washed three times with PBS and fixed in methanol for 5 min. The cells were lastly stained with 1% crystal violet (Beyotime) for 15 min and washed with PBS. Images were collected using a scanning apparatus (Canon, Tokyo, Japan).

### 4.6. Western Blot Analysis

Briefly, MDA-MB-231 cells were lysed in lysis buffer (50 mM Tris-HCl pH 8.0, 5 mM EDTA, 100 mM NaCl, 0.5% NP-40, 1 mM PMSF) and then centrifuged for 10 min at 12,000 r, and the insoluble debris was discarded. Cell lysates were further analyzed with SDS-PAGE and western blotting with indicated antibodies. After treatment with **13a** and **13c** for indicated times, cells were harvested with RIPA buffer (150 mM sodium chloride, 1% Triton X-100, 0.5% sodium deoxycholate, 0.1% SDS, 50 mM Tris and cocktails of protease and phosphatase inhibitors) for 10 min at room temperature and boiled for another 10 min. Equal amounts of total proteins (35 μg) underwent 15% SDS-PAGE and were electroblotted onto the polyvinylidene difluoride (PVDF) membrane. The membrane was blocked with 5% (*w*/*v*) fat-free dry milk in PBS-Tween 20 (PBST; 0.05%) for 1 h and incubated with primary antibody (1:1000 in PBST) at 4 °C overnight. After three washings in PBST, the PVDF membrane was incubated with appropriate horseradish peroxidase-conjugated secondary antibodies (1:20,000) for 1 h at room temperature. The immunoreactive bands were developed with the ECL western blotting system. β-Actin was used as loading control. The relative quantity of proteins was analyzed via the Image J software (NIH, Bethesda, MD, USA).

### 4.7. Anti-Migration Assay

MDA-MB-231 cells were seeded in a six-well plate to form a monolayer cell tile state. A 200 µL pipette tip was used to create a linear wound. The wound was washed with PBS to remove damaged cells and then recorded under a microscope. Compounds **13a**, **13c** and SAHA were then added, and the wells were further photographed at 24 and 48 h, respectively. Results were expressed as the percent of wound healing that is, the distance migrated at 0 h minus the distance migrated at 24 or 48 h relative to the distance migrated at 0 h.

### 4.8. Cell Cycle Arrest Analysis and Apoptosis Analysis

Briefly, cell cycle analysis was carried out by estimating DNA contents with flow cytometry at 488 nm. MDA-MB-231 cells (4 × 10^4^) were incubated in a small dish for 12 h and then medicated. After 24 h of cell cycle treatment, the cells were washed three times with PBS, and the cell suspension was collected, centrifuged again, and then added to an ethanol fixative (70%) and placed in a refrigerator for overnight storage. After washing, the dye was added and stained with PI containing RNaseA solution for 30 min at 37 °C, and then analyzed by FACS.

Briefly, cell apoptosis analysis was measured by annexin V FITC/PI assay using Annexin v-PE/7-AAD Apoptosis Detection kit (BD). MDA-MB-231 cells (2 × 10^5^) were incubated in a six-well plate for 12 h and then medicated. After 48 h, the drug-treated cells were washed three times with PBS; the cell suspension was collected and centrifuged; and the supernatant was removed. Finally, the corresponding buffer solution (500 μL) and the dye (5 μL) were added to each sample, incubated at room temperature for 15–20 min and then detected by flow cytometer at 488 nm.

### 4.9. Molecular Docking Studies

Molecular docking studies were carried out with Autodock-4.27. For the docking calculations HDAC1 crystal structure (PDB code: 4BKX) was retrieved from the Protein Data Bank (www.pdb.org). For protein preparation, all the water molecules were removed from HDAC1, and Gasteiger partial charges were assigned to the selected compound and enzyme atoms. The docking results were analyzed with the programs AutoDockTools (Olson, CA, USA), 27 DOCKRES and VMD.

## 5. Conclusions

We designed and synthesized a new series of coumarin-based hydroxamate HDACis and evaluated their biological activities in a series of in vitro assays. Most compounds showed excellent HDAC1 inhibitory activities and also displayed significant growth inhibition against different human cancer cells. Among them, compounds **13a** and **13c** were two to three times more active than SAHA, and further experimental investigations were conducted. Immunoblot analysis revealed that **13a** and **13c** dose-dependently increased the acetylation of histone H3 and H4, confirming their HDAC1 inhibitory effects. Furthermore, cell migration and colony formation assays showed that the two compounds displayed anti-metastatic and anti-proliferative activities. Moreover, **13a** and **13c** arrested MDA-MB-231 cells at G2/M phase and induced cell apoptosis. Finally, a molecular modeling study was also performed to assess the potential binding ability of **13c** with HDAC1. Together, the present work afforded new HDACis that could be further investigated as promising anticancer candidates.

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
