# Peer review of "Development of Coumarin-Based Hydroxamates as Histone Deacetylase Inhibitors with Antitumor Activities"

_molecules, 2020, doi:10.3390/molecules25030717_

Round 1

Reviewer 1 Report

Title: Development of coumarin-based hydroxamates as histone deacetylase inhibitors with anti-tumor activity

Summary: The authors tried to synthesize new series of coumarine-based hydroxamate HDAC inhibitor. Through this research, the authors developed more potent HDAC1 inhibitor compared to vorinostat, which has been approved from FDA, but I have the following queries and comments.

Comments

First of all, the authors need to clarify the reason for compound selection in Section 3.2.

It would be more informative to describe about correlation between potency and moiety of HDAC inhibitors.

Did the selected HDACi derivatives have HDAC1 specific inhibition effect?

In validation experiments, the concentration of 13a and 13c were different in figure3-6. Although the anti-proliferation and migratory effect of 13a and 13c was exhibited in sub-dose (0.25-1uM), acetylation status of H3 and H4 was revealed in high dose (2-5uM). Based on these results, the possibility that anti-proliferation and migratory effect as well as cell cycle arrest of 13a and c resulted in off-target effect, is hard to be excluded. The authors should clarify this issue.

Furthermore, the overall flow of this manuscript, especially results, is not strict forward and results are overstated with no enough supporting data. The results part should be re-organized.

Author Response

Dear Editor Lola Huo:

We sincerely thank you and the reviewers for the constructive suggestions of our manuscript entitled “Development of coumarin-based hydroxamates as histone deacetylase inhibitors with antitumor activities” (ID: molecules-694023) and the opportunity to submit the revised version. In the revised manuscript, we have made all the changes requested by the reviewers and the revisions have been highlighted in yellow. Therefore, we believe that the quality of the manuscript have been significantly improved and qualified for publication in Molecules. Please find below a point-to-point response to the reviewers' comments.

Reviewer 1:

1)First of all, the authors need to clarify the reason for compound selection in Section 3.2. It would be more informative to describe about correlation between potency and moiety of HDAC inhibitors.

Response: Thanks for the reviewer’s comments. According to the SAR analysis in Section 3.1, we have found that in the alkoxy substituted coumarin derivatives (13-series), the proper length and shape of the alkyl chain were very important to the HDAC1 inhibitory activity, and in benzyloxy substituted ones (14-series), compounds with fluoro-substitution at the ortho-position of benzyl ring showed better enzyme inhibitory activities than other analogues. Especially compounds 13a-d and 14m displayed much better HDAC1 inhibitory activities (IC50 £ 0.50 nM) than the control drug SAHA (IC50 = 12.36 nM). Then we selected these compounds for further detection. We have added extra descriptions regarding this point to Section 3.2 and highlighted them in yellow in the revised manuscript.

2)Did the selected HDACi derivatives have HDAC1 specific inhibition effect?

Response: Thanks for the reviewer’s comment. As described in our previous report (Molecules 2019, 24(14), 2569; doi: 10.3390/molecules24142569), we found that the coumarin-based hydroxamate HDACis were pan-HDACis which were similar to SAHA. The selected HDACis in this manuscript were derivatives from the previous structures, so we could assume that they were not specific HDAC1 inhibitors but were only more active against HDAC1 than other subtypes.

3)In validation experiments, the concentration of 13a and 13c were different in figures 3-6. Although the anti-proliferation and migratory effect of 13a and 13c was exhibited in sub-dose (0.25-1 uM), acetylation status of H3 and H4 was revealed in high dose (2-5 uM). Based on these results, the possibility that anti-proliferation and migratory effect as well as cell cycle arrest of 13a and c resulted in off-target effect, is hard to be excluded. The authors should clarify this issue.

Response: Thanks for the reviewer’s comment. Actually in the initial experiment, we have explored the effect of compound 13a on the acetylation level of histone H3 in lower dose range (0.5-2 mM, see figure below). The results revealed that 13a could also obviously increase the acetylation level of histone H3, while the increasing trend of the acetylated protein level from 0.5 to 1 mM was not obvious. Therefore, in order to see a better dose-dependent trend, we increased the concentration range of tested compounds.

Meanwhile, these compounds are similar structures to SAHA but just different subtype, and they belong to the same series of analogues as those in the previous report (Molecules 2019, 24(14), 2569; doi: 10.3390/molecules24142569). Thus their antitumor effects are definitely caused by targeting HDACs, which was also supported by their non-cytotoxicity against the normal cells as described in the manuscript.

4)Furthermore, the overall flow of this manuscript, especially results, is not strict forward and results are overstated with no enough supporting data. The results part should be re-organized.

Response: Thanks for the reviewer’s comment. We have reorganized the results part of the article. Relevant changes have been highlighted in yellow.

Reviewer 2 Report

In the manuscript named "Development of coumarin-based hydroxamates as histone deacetylase inhibitors with antitumor activities," the collective of authors Na Zhao et al. are presenting newly designed and synthesized coumarin-based hydroxamate HDAC inhibitors. And, evaluates their biological activities by using several in vitro assays. The authors proposed novel compounds that act as an excellent HDAC1i that induce cell cycle arrest and early-stage apoptosis in several different cancer cell lines.
After reading the manuscript, I have some major as well as minor points that the authors should address.

Major
Regarding the Figures should be larger/bigger to be more readable for readers — especially those related to the Results and discussion part.
Also, Figures should be described in more detail in the text, as well as the Legends, which should be extended with relevant information.
Since Journal Molecules has no limitation in the length of the submitted manuscripts, I would strongly advise the authors to extend the Material and Method part of the paper. This passage is particularly concise without any specification of used antibodies, instrumentation, or statistics.
Minor:
row 40: misspelling panobinostat instead of panorbistat
row 74: Scheme 1. should by underneath the picture

In this stage, I can not recommend the acceptance of the manuscript for publishing.

Author Response

Dear Editor Lola Huo:

We sincerely thank you and the reviewers for the constructive suggestions of our manuscript entitled “Development of coumarin-based hydroxamates as histone deacetylase inhibitors with antitumor activities” (ID: molecules-694023) and the opportunity to submit the revised version. In the revised manuscript, we have made all the changes requested by the reviewers and the revisions have been highlighted in yellow. Therefore, we believe that the quality of the manuscript have been significantly improved and qualified for publication in Molecules. Please find below a point-to-point response to the reviewers' comments.

Reviewer 2:

Major points:

1)Regarding the Figures should be larger/bigger to be more readable for readers especially those related to the Results and discussion part.

Response: Thanks for the reviewer’s suggestion. We have enlarged all the figures in the revised manuscript as suggested

2)Figures should be described in more detail in the text, as well as the Legends, which should be extended with relevant information.

Response: Thanks for the reviewer’s constructive suggestion. We have extended the legends of relevant figures with more information and added more details of results where necessary.

3)Since Journal Molecules has no limitation in the length of the submitted manuscripts, I would strongly advise the authors to extend the Material and Method part of the paper. This passage is particularly concise without any specification of used antibodies, instrumentation, or statistics.

Response: Thanks for the reviewer’s constructive suggestion. We have added relevant information in the "Materials and Methods" part of the paper. Also more details of the experimental procedures including the specification of used antibodies, instrumentation, etc., in section 4.1, 4.5, 4.6, 4.7 and 4.8 as highlighted in the revised manuscript

Minor points:

row 40: misspelling “ panobinostat ” instead of “ panorbistat ”

row 74: Scheme 1. should by underneath the picture

Response: Thanks for the reviewer’s comment. We have corrected “panorbistat” to “panobinostat” in the revised manuscript (Line 40). Also we have moved the text of “Scheme 1” underneath the picture as suggested.

We have also corrected a few minor language slips and misspellings in the revised manuscript.

Round 2

Reviewer 1 Report

The authors have answered all my comments. No further questions.